# Spatial organization in microbial range expansion emerges from trophic dependencies and successful lineages

Benedict Borer [1✉], Davide Ciccarese [1,2], David Johnson [2] & Dani Or [1]

Evidence suggests that bacterial community spatial organization affects their ecological function, yet details of the mechanisms that promote spatial patterns remain difficult to resolve experimentally. In contrast to bacterial communities in liquid cultures, surface-attached range expansion fosters genetic segregation of the growing population with preferential access to nutrients and reduced mechanical restrictions for cells at the expanding periphery. Here we elucidate how localized conditions in cross-feeding bacterial communities shape community spatial organization. We combine experiments with an individual based mathematical model to resolve how trophic dependencies affect localized growth rates and nucleate successful cell lineages. The model tracks individual cell lineages and attributes these with trophic dependencies that promote counterintuitive reproductive advantages and result in lasting influences on the community structure, and potentially, on its functioning. We examine persistence of lucky lineages in structured habitats where expansion is interrupted by physical obstacles to gain insights into patterns in porous domains.

[1] Department of Environmental Systems Science, ETH Zürich, 8092 Zürich, Switzerland. [2] Department of Environmental Microbiology, Swiss Federal Institute of Aquatic Science and Technology (Eawag), 8600 Dübendorf, Switzerland. ✉email: benedict.borer@usys.ethz.ch

Bacterial life in natural environments is predominantly associated with sessile assemblages of cells that form aggregates or surface-attached biofilms[1–5]. The confinement of bacterial cells in sessile colonies and their limited relocation opportunities makes positioning in space a critical factor for access to nutrients[6], exchange of metabolites[7], protection from phage infection[8], or protection from predation[9]. Within such highly constrained systems, scenarios emerge in which a few cells contribute disproportionally to the long term community biomass[10] supporting the notion of "survival of the luckiest". Evidence in the form of genetic segregation of initially mixed communities is observed frequently in colonies growing on surfaces[11–18], or observations of detrimental mutations accumulating at the assemblage periphery that are rapidly lost in homogeneous environments such as liquid cultures[19]. During sessile growth, bacterial cells are essentially fixed in space and depend on diffusional fluxes dictated by heterogeneous nutrient landscapes (governed by the surface and their neighbors) that ultimately shape the resulting bacterial assemblage[6]. In contrast to advection dominated planktonic life style, bacterial cells in diffusion dominated systems often self-engineer their immediate surroundings (a process frequently termed niche construction[20]). For example, single species biofilms engage in cross-feeding interactions through divergent behavior of spatially segregated subpopulations experiencing fundamentally different growth conditions[21,22]. Overall, trophic interactions have the potential to shape the emerging community patterns during bacterial range expansion. When competing for the same resource, the front of an expanding population offers advantages due to preferential nutrient access and reduced competition that permit the unhindered expansion of individual cell lineages, forming genetically segregated sectors[11]. In contrast, cross-feeding interactions, where one species relies on nutrients or modification of its imminent surroundings provided by another species within the assemblage[23] add constraints by the need for close proximity to the interacting partner. The signature of such interactions is seen in the spatial organization of cells within the assemblage[24–27] with the potential of sequential expansion if trophic interactions are unilateral[28]. Interestingly, these sequential patterns can be interrupted by a second pattern[28], where large monoclonal sectors emerge due to rare nucleation events (where monoclonal sectors are seeded due to a combination stochastic but mostly deterministic conditions). These patterns mediate various aspects of community functioning and genetic makeup; the disturbance or absence of spatial structure may adversely impact community ecosystem functions[7,29,30]. Experimentally resolving the mechanisms that give rise to different community expansion patterns remain a challenge due to difficulties in visualizing and tracking individual cells, and limitations in representing the underlying and highly dynamic nutrient landscape. Turning to mathematical models that simulate both the nutrient conditions and bacterial processes at the relevant scale to individual cells provides an opportunity to unveil mechanisms giving rise to experimentally observed community patterns at the macro scale. We hypothesize that the spatial organization of sessile, cross-feeding bacterial assemblages undergoing range expansion is shaped by highly localized and dynamic differences in cell growth rates that are reinforced by feedbacks, or self-engineered of local nutrient landscape. To systematically and quantitatively evaluate the hypothesis and the mechanisms postulated, we developed an agent-based model that combines Monod type growth kinetics with a simple mechanical shoving algorithm for bacterial growth in homogeneous and structured habitats similar to experimental setups used for independent validation. The detailed description of chemical and metabolite diffusion at high spatial resolution links localized nutrient conditions and spatially variable growth

rates as a key driver that promote community pattern formation. Finally, we discuss the influence of obstacles such as found in structured habitats (porous media or rough surfaces) on the spatial self-organization, community proliferation, and genetic segregation. Our modelling framework provides novel mechanistic insights into experimental observations of spatial self-organization during bacterial range expansion, predicting a feedback between the emerging colony pattern and self-engineered nutrient landscape at a resolution currently inaccessible to experimental quantification. These insights are used to visualize the importance of the spatial dimension in shaping the community composition concerning individual "lucky" bacterial lineages and potential ramification for evolutionary mechanisms.

## Results

**Dynamic nutrient landscapes mediate colony spatial self-organization.** We use a synthetic community containing two isogenic mutants of *P. stutzeri* A1501[28,31] cross-feeding nitrite in the denitrification pathway (Fig. 1a) to systematically investigate mechanisms, that shape the spatial self-organization of trophically interacting bacterial colonies during range expansion. The inherent toxicity of nitrite at low ambient pH modify the type of trophic interactions from competition (complete reducer growing anaerobically on nitrate, negative bilateral), via weak mutualism (producer and consumer cross-feeding nitrite anaerobically at pH 7.5, positive unilateral) to strong mutualism (producer and consumer cross-feeding nitrite anaerobically at pH 6.5, positive bilateral). Consequently, we observe experimentally two fundamentally different expansion mechanisms depending on the type of trophic interactions. Competition for the same resource fosters segregation and the formation of well-defined sectors of each bacterial strain (Fig. 1b). Similar patterns have been observed frequently in other studies[11,28,31]. The onset of strong mutualistic interactions promote an increased intermixing of the two strains (Fig. 1c) that we attribute to the need for reducing diffusion distances between interacting partners[30,31]. Weak mutualistic growth invokes two distinct spatial patterns: a sequential pattern where producer cells dominate the colony periphery[28,31], and in certain rare cases where large consumer branches emerge at the colony periphery (Fig. 1d). The dynamics of super-sectors (defined as the rare but large monoclonal sectors observed in weak mutualistic scenarios contributing disproportionally to colony expansion) are difficult to investigate experimentally and require insights into the spatially variable growth rates and shoving mechanisms.

The mathematical model reveals strong links between the underlying nutrient landscape and localized growth rates that shape the colony patterns (Fig. 2). When observing the nutrient conditions in the proximity of the expanding colony edge (indicated by the red rim in Fig. 2d), both complete reducer strains experience the same nitrate concentrations and hence equal growth rates as they are essentially isogenic strains in the competitive scenario (Fig. 2a). Strong mutualism (nitrite being toxic, pH 6.5) results in a lower but similar growth rate (converging) of the two bacterial strains, hence, smaller colony size for the same incubation time when compared to the competition scenario. A significantly different pattern characterizes the weak mutualistic consortium. After an initial lag phase due to nitrite unavailability, the consumer strain benefits from an overall higher growth rate compared to the producer strain in the weak mutualistic scenario (Fig. 2a). This mechanism is rooted in the underlying nutrient landscape (Fig. 2b, c). At pH 6.5, nitrite toxicity controls the proliferation of the consumer strain and equal production and consumption keep nitrite concentrations low. At pH 7.5, nitrite is nontoxic and begins to accumulate

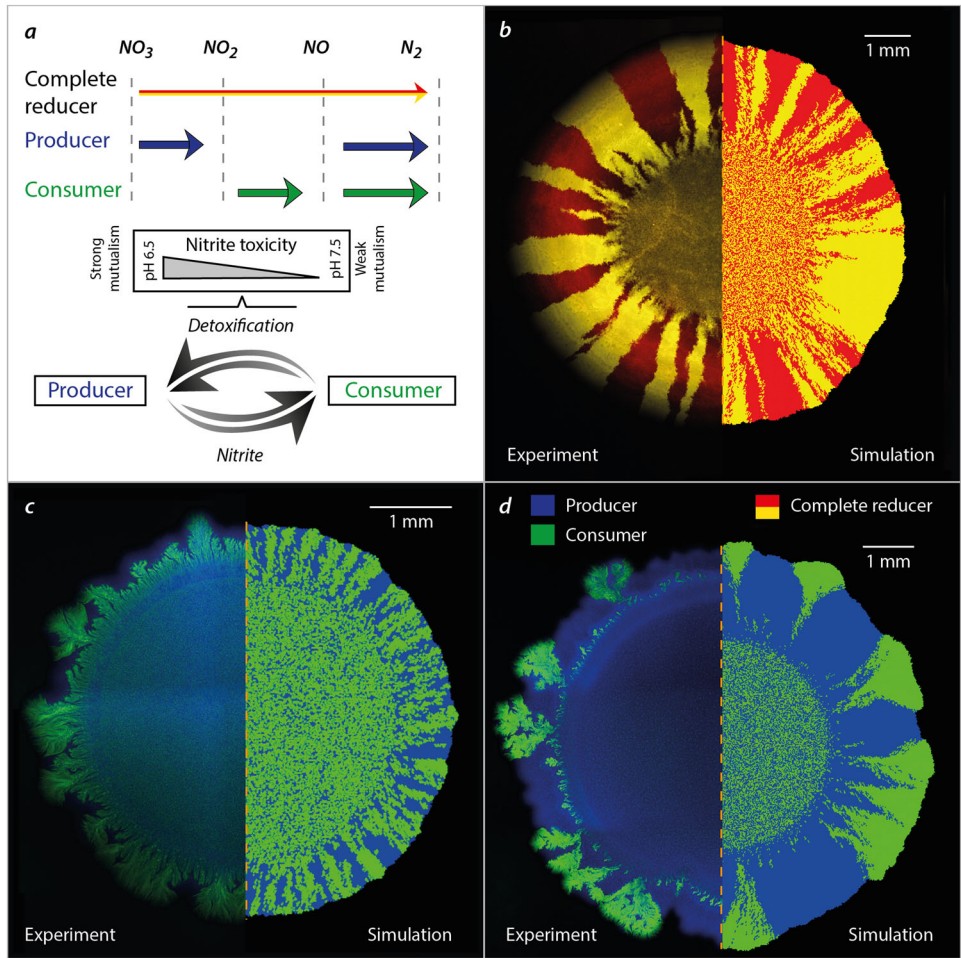

**Fig. 1 Spatial patterns emerging from different trophic interactions among members of a synthetic bacterial community grown on agar surfaces. a** The denitrification metabolic pathway of the complete reducer and the producer and consumer used in this study. The complete reducer *P. stutzeri* A1501 employs the entire denitrification pathway, reducing nitrate ($NO_3^-$) all the way to dinitrogen gas ($N_2$). Two nitrite cross-feeders derived from the complete reducer were used to form a mutualistic consortium: (i) a producer converts nitrate to nitrite ($NO_2^-$) but is unable to complete the denitrification pathway due to a lack of nitrite reductase; (ii) a consumer unable to use nitrate directly as an electron acceptor (lacking the enzyme nitrate reductase) and thus depends on the producer supplying nitrite. The toxicity of the intermediate metabolite nitrite promotes a mutualistic relationship where the strength of the interaction is controllable due to the variable toxicity of nitrite depending on local pH (acidity). **b** Comparison of experimental and simulation data of a competitive scenario represented by two complete reducer *P. stutzeri* strains. Simultaneous expansion of the two isogenic but fluorescently tagged strains creates visible sectors of strains tagged with the same fluorescence. **c** Comparison of experimental and simulation data of a mutualistic interaction, where producer is tagged with CFP (blue) and consumer with GFP (green). Due to the higher toxicity of nitrite the community is more intermixed. **d** Comparison of experimental and simulation data of a weak mutualistic (pH 7.5) scenario where producer is tagged with *ecfp* (blue) and consumer with *egfp* (green). The colony follows sequential expansion lead by the producer with visible sectors of consumers emerging.

during the early stages of colony growth due to the sequential expansion of the producer followed by the consumer. In some rare cases, consumers are pushed ahead by the proliferating producers (Fig. 2d) and remain in the actively growing colony periphery. Once sufficient nitrite is available, these consumer cells residing at the periphery proliferate disproportionally due to their relative growth rate advantage and create large monoclonal consumer sectors. This process, remaining at the periphery through passive motion until favorable conditions enable emergence of super sectors, is further defined as "nucleation of super-sectors" for simplicity. These differences become apparent when visualizing dynamic growth rates of individual agents during the simulation depending on the location and strain (Fig. 2d). Producer cells remaining at the colony periphery retain high growth rates throughout the simulation slightly declining due to nitrate diffusive limitations. Consumer cells on the other hand have an initial disadvantage owing to the lack of nitrite,

which first needs to be provided by the producer cells. Thus, consumer cells are reliant on proliferation and shoving from producer cells to stay at the colony periphery. Once the surrounding producers provide sufficient nitrite, the cells which managed to remain at the colony periphery proliferate into large sectors of kin cells due to a relative growth rate advantage compared to the surrounding producers (no substrate limitation). These observations highlight how stochastic effects (shoving of the consumer cells through producer proliferation) and deterministic processes (higher relative growth rate of the relocated consumer cells resulting in concave sector boundaries) manage to govern the two different observed patterns in weak mutualistic conditions.

**Lineage tracking reveals lucky individuals and spatial genetic bottlenecks.** The mechanistic model offers opportunities that are not yet possible experimentally, namely to track lineages of

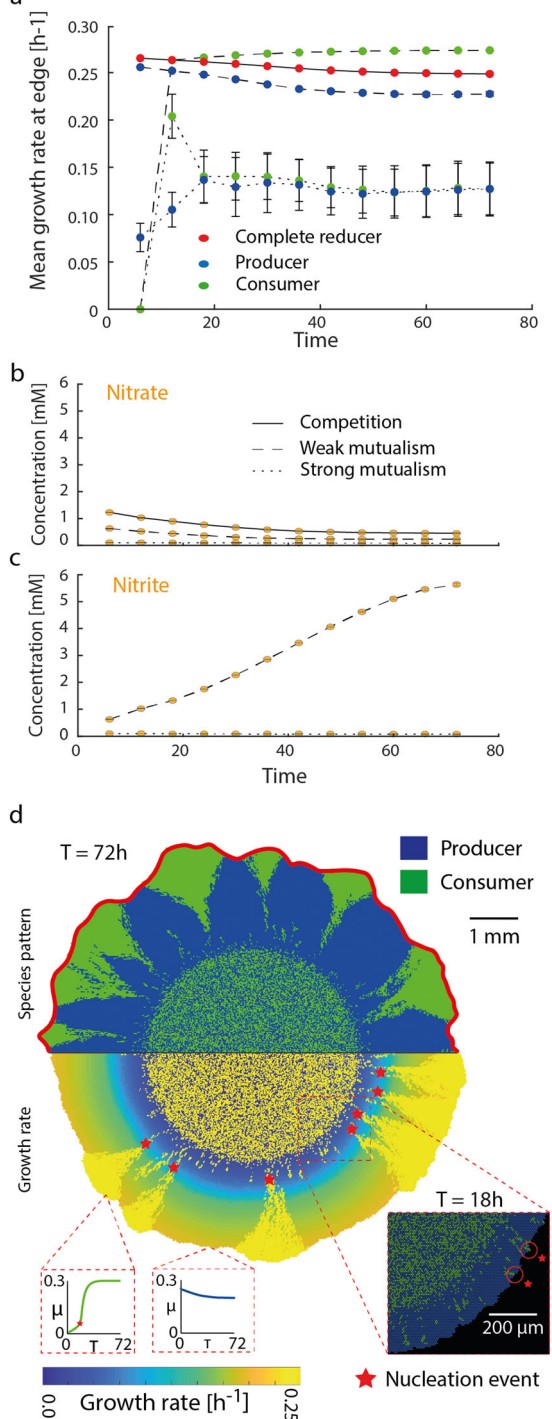

**Fig. 2 The underlying chemical landscape governs local growth rates of the bacterial colony.** Nutrient concentrations and growth rates within the growth layer for all trophic interactions (red zone in **c**). **a** In the competitive scenario, predicted growth rates are equal for the complete reducers having the same kinetic parameters. Weak mutualism is characterized by a relative growth rate advantage due to the ubiquitous nitrite availability. In mutualistic scenario, both strains have a lower but equal growth rate due to nitrite toxicity, emphasizing their mutualistic interaction. Dynamics of nitrate (**b**) and nitrite (**c**) for all trophic interactions. Competition results in a slow decline of nitrate with no nitrite produced (complete reducer). Weak mutualistic conditions promote the accumulation of nitrite as there is negligible toxicity in this condition. Strong mutualism results in a rapid decline of nitrate (toxicity acts on the yield), with overall low nitrite levels (removed by consumer cells in close proximity). **d** Spatial distribution of the two species community in weak mutualistic conditions (top) in relation to the predicted growth rates (bottom) at the end of the simulation time. Following individual cells within the simulation reveals different growth rate histories depending on the strain and location. Growth rates of producer cells are dictated by nitrate availability and proximity to the colony periphery (due to nitrate diffusion). Consumer cells initially cannot grow due to nitrite limitation and subsequently proliferate if remaining sufficiently close to the colony periphery through shoving. Rare nucleation events indicate the creation of large monoclonal sectors that arise due to favorable spatial positioning (initially shoved by the producer cells) and nutrient conditions.

bacterial "virtual" cells within the complex and spatially varying community. The assumption is that variations in reproductive success of the ancestral population (inoculated cells) results in some cell lineages (defined as the ancestral inoculated cell and all of its progeny), that contribute disproportionally to the final colony biomass as depicted as pseudo-colors in Fig. 3. For competitive trophic interactions, the model predicts a clear segregation of the two strains into sectors that prominently persist into the colony periphery (as also observed experimentally shown in Fig. 1a and in other studies). An important observation that could not be resolved in the experiment is that a single sector containing the same strain does not consist of a single ancestor (i.e., not originating from the same inoculated cell). Strong

mutualism favors intermixing of the two strains with considerably thinner sectors compared to the trophic competition scenario. Sequential expansion as observed under weak mutualistic interactions alters the observed pattern. For the majority of the colony, the producer expands radially (perpendicular to the colony periphery) with no penetration of consumer sectors similar to the pattern observed in the competitive scenario (essentially equal patterns since producers compete for nitrate). Super sectors that penetrate the layer of producers are typically monoclonal, owing to the fact that they typically nucleate from a single cell (ratio of consumer lineages at the periphery to consumer branches is 1.25 ± 0.05). In between consumer branches, an increased rate of coalescence of producer lineages can be observed that arise from spatial bottlenecks imposed by the flanking consumer sectors. The degree to which a community de-mixes during range expansion experiments is commonly quantified by counting the number of transitions between different strain sectors at the colony periphery, and can be tracked dynamically in the simulation (Fig. 3b, legend in Fig. 3c). For competitive trophic interactions, the total visible sectors remain stable throughout the simulation with a slight negative trend due to coalescence events, where individual sectors are pinched and buried behind the colony expansion front. Sequential expansion of two strains promotes rapid demixing for weak mutualistic interactions, with some more branches resurging in the later stages of the simulation.

For the strong mutualistic interactions scenario, the tight association between the two strains favors intermixing and thus an overall higher number of individual branches is seen at the end of the simulations. The modelling results follow capabilities of the experimental system, where distinction between sectors can only be made based on fluorescent differentiation. By quantifying the unique lineages that proliferate to the colony periphery, differences in demixing dynamics between the strains become evident. When competing for the same resource, the two strains demix very rapidly, as they are essentially equivalent except for the fluorescent label (Fig. 3c). In the case of strong mutualism, demixing of both strains is slower compared to the competitive

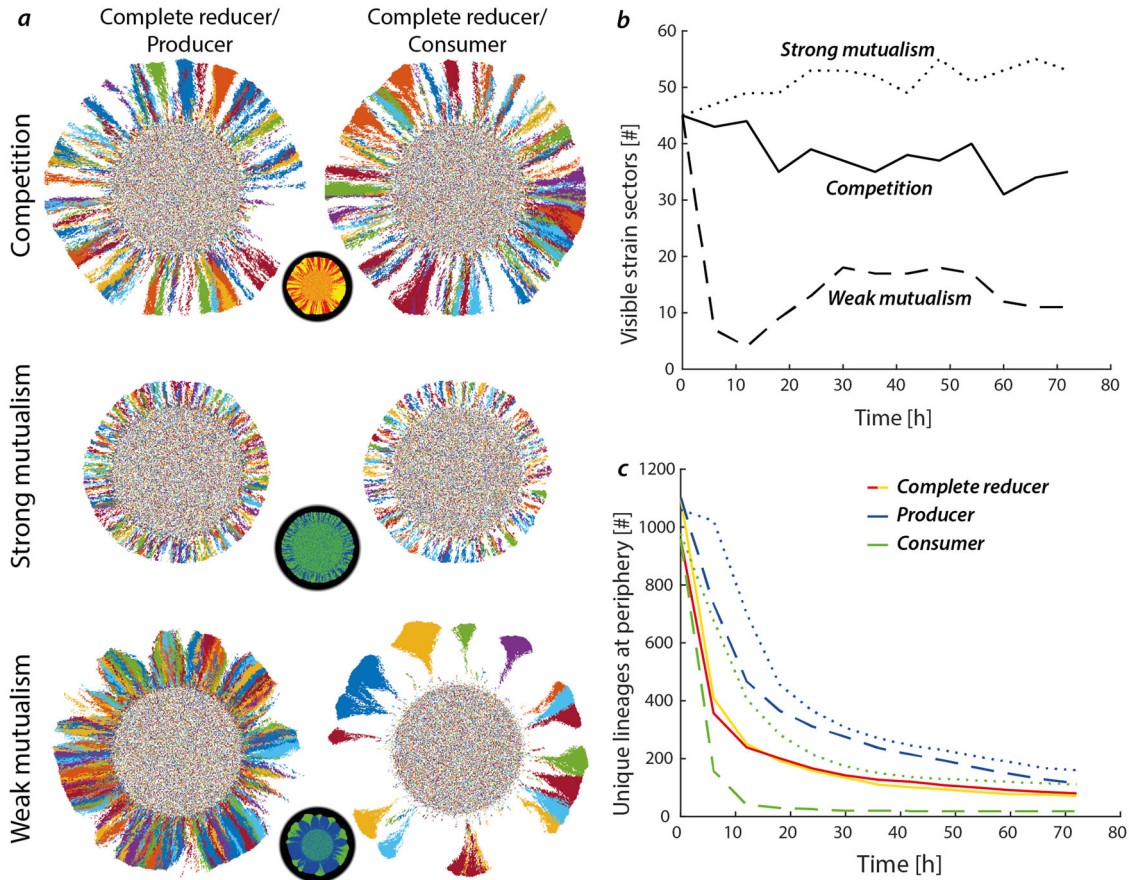

**Fig. 3 Trophic interactions alter the reproductive success of individual cells. a** Reproductive success of individual lineages with associated spatial patterns of demixing depending on trophic interaction. Competition results in well-defined sectors where one sector typically contains multiple ancestral lineages. Strong mutualism favors intermixing of the two strains resulting thin sectors. In case of weak mutualism, the producer strain primarily engages in competitive patterns due to nutrient competition with its kin. Large consumer sectors emerge where individual ancestral lineages become dominant and produce monoclonal sectors. **b** The number of interspecies boundaries remains stable in the competitive scenario. In weak mutualistic scenario, the number of interspecies boundaries shows a decline due to the loss of most consumer branches. An increase in the number of interspecies boundaries in the strong mutualistic case is attributed to branches bifurcating and thus increased intermixing of the two strains. **c** The total number of unique lineages residing at the colony periphery shows a steep decrease during early colony development asymptotically reaching steady state once sectors have developed. The decrease follows different patterns depending on trophic interactions mediated by the varying mechanisms of range expansion (sequential vs. simultaneous). Each scenario was replicated six times.

case where more individual lineages manage to proliferate at the colony periphery in the case of the consumer strain. Weak mutualism promotes a strong demixing of the consumer strain whereas the producer has a slower demixing with time compared to the competitive scenario, since more space is available initially lacking growth by the consumer until sufficient nitrite has accumulated. Overall, from the inoculated cells residing within the actively growing layer at the colony periphery, approximately 80% of all cell lineages are lost during range expansion which increases to 99% when taking into account all inoculated cells (i.e., cells at the colony center). The loss of lineages during colony expansion underlines the variation in reproductive success of individual lineages, where a fraction of the initial inoculum (0.31% ± 0.02%, 0.32% ± 0.03%, and 4.96% ± 0.06% for the competitive, weak mutualistic, and strong mutualistic scenario, respectively) contributes more than 50% of the final community biomass.

**Spatial structure mediates self-organization with genetic consequences.** Definitive studies for disentangling the effects of structured environments (porous media or rough surfaces) from those induced by trophic interactions on the resulting spatial

organization and genetic lineages are rare. We have used the model to simulate, for identical trophic scenarios (competition, weak mutualism, and strong mutualism) the resulting community patterns expanding in structured environments with varying density of obstacles simulated as solid particles (Fig. 4a–c). An increase in obstacle density results in a higher genetic segregation of the two strains when competing for the same nutrient (Fig. 4a, competition) with fewer individual cell lineages persisting to the colony periphery (Fig. 4d). A similar effect is expected for the producer strain in weak mutualistic conditions, as they are equally dependent on the diffusion of nitrate from the periphery. The increase in physical obstacles has a different effect on the consumer strain branches in the weak mutualistic case (Fig. 4b). Although a similar number of monoclonal consumer branches emerge in the early stages of colony expansion, the obstacles hindered the proliferation of these branches resulting in a limited number persisting to the colony periphery in medium and fine structured habitats (Fig. 4e). Interestingly, more consumer branches persist to the periphery in coarse conditions compared to a homogeneous habitat. Emerging consumer branches are pushed through bottlenecks by surrounding producer cells, which facilitates their persistence to the periphery. In contrast to the

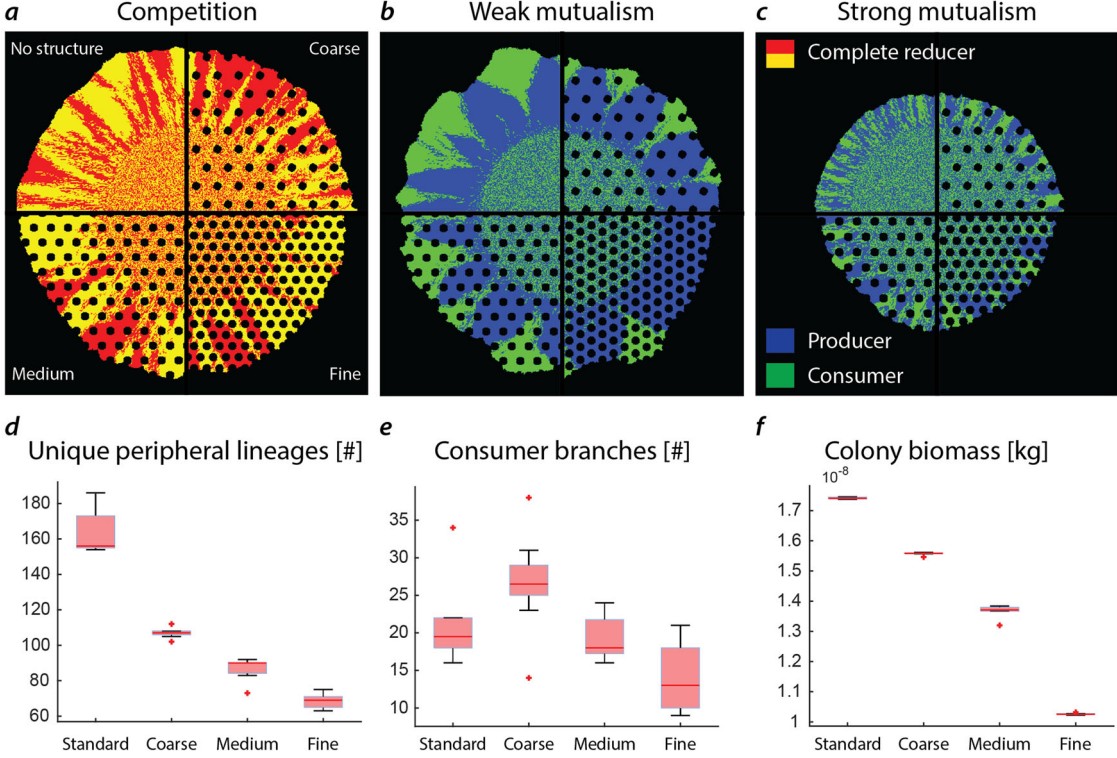

**Fig. 4 Simulated two-species bacterial colony expansion into domains with different spatial solid structures—impacts on lineage proliferation with potential ecological and evolutionary consequences.** Visualization of the influence of a structured environment on the emerging colony pattern for competing (**a**), weak mutualism (**b**), and strong mutualism (**c**) in relation to the structure density. **d** When competing, the addition of structures reduces the number of cell lineages proliferating to the periphery due to trapping of cell lineages in front of the obstacle. This leads to greater genetic segregation of the community and fewer lineages persisting to the assemblage periphery. **e** Weak mutualistic conditions result in a similar picture concerning producers. Consumer branches that nucleate in front of an obstacle are limited in their persistence due to spatial constrains and the total number of proliferating branches decrease with increasing obstacle density changing the ecological functioning of the assemblage. Interestingly, the introduction of coarse obstacles to a homogeneous habitat increases the number of consumer branches (although statistically not significant, $p$-value of 0.12, two-sample $t$-test). **f** In mutualistic conditions, the increased segregation of the two strains is detrimental to the function of the whole community due to the increased diffusion distance between the strains resulting in an overall lower expansion rate of the community. Quantification of the unique peripheral lineages, peripheral consumer branches, and biomass is summarized in SI Table 1. Six replicates were simulated for scenarios void of obstacles. For each scenario containing obstacles, ten replicates were simulated.

medium and fine structured habitats, these branches do not encounter many additional obstacles that may inhibit their proliferation. Strong mutualistic conditions favor a higher degree of intermixing resulting in shorter average diffusion lengths (Fig. 4c). Including obstacles in the simulation results in decreased intermixing of the two strains and thus longer average diffusion lengths expressed in an overall lower total biomass created during colony growth (Fig. 4f).

## Discussion
It has been frequently documented that changes in trophic interactions invoke fundamentally different spatial self-organization and expansion mechanisms such as simultaneous expansion when competing for the same resource[11,28,31], simultaneous expansion at higher intermixing ratios during strong mutualistic growth[30–32] or sequential expansion for cross-feeding an intermediate metabolite with positive unilateral of the interacting species[28]. In the latter case, two significantly different patterns are observed within the same colony: (i) sequential expansion where the producer strain advances first and the consumer strain follows and (ii) localized super-sectors of the consumer strain having a higher growth rate compared to the surrounding producer strain[28,31]. Important questions concerning these super-sectors are the mechanisms that promote the emergence of the super sectors and their persistence at the colony

periphery. In order for super sectors to emerge, two conditions are required: an initial phase where consumers manage to stay at the periphery despite a growth rate disadvantage and a subsequent phase, where consumers have a growth rate advantage indicated by the curved sector boundaries. In the simulations, growing producer cells initially shove consumers located at the colony periphery until sufficient nitrite is produced for them to proliferate.

An in-depth experimental evaluation of the importance of initial spatial positioning, and the role of cell shoving is complex due to massive amount of replicates and inability to remove cell shoving in a living community. Alternatively, we have removed the shoving component in the mathematical model (i.e., allowing only peripheral cells contribution to colony expansion through placement of their progeny on unoccupied grid nodes). The result was an overall lower expansion of the colonies (SI Fig. 1a–c) with conspicuous omission of super sectors in the weak mutualistic scenario (SI Fig. 1b). Additionally, we could easily control the initial positioning of consumers and producers in the mathematical model. Encasing the colony with consumer cells resulted in myriads of consumer super sectors (SI Fig. 1d), whereas encasing the inoculated colony with producer cells suppressed the emergence of any consumer super sectors as they are buried behind the advancing producer front. These additional simulations highlight the importance of initial spatial positioning and

cell shoving for the seeding of the observed consumer super sectors, in case of the weak mutualistic scenario.

The mechanism enabling the consumer individuals to proliferate disproportionate compared to the producer at the colony periphery is embedded within the underlying nutrient landscape, where localized relative growth rate differences between the two strains emerge dynamically (Fig. 2d). In contrast to the producer, the consumer strain is not dependent on diffusion of peripheral nitrate and has a ubiquitous supply of nitrite provided by the more abundant producer, resulting in a higher growth rate relative to the producer in the later stages of colony expansion (Fig. 2b, c). In comparison to other processes that result in spatially self-organized patterns such as the classical Turing pattern processes[33] or phase separation[34], the spatial organization in our model is governed by localized growth conditions, self-engineered metabolic landscapes and favorable initial spatial positioning that differ fundamentally from the system properties in both the above-mentioned theories. The shape of the emerging super sector has previously been linked to the relative growth rate advantages[35]. Similar observations have been made in studies focusing on the mechanical forces shaping the spatial patterns of mutant strains with relative growth rate differences during range expansion[17,18]. In both cases, cells having a growth rate disadvantage managed to stay at the colony periphery through mechanical interactions, and shoving of the surrounding faster growing cells. Although the resulting colony pattern in our system is similar to the above-mentioned studies, where growth rate differences are imposed via mutations and are thus fixed in time, the mechanism is fundamentally different. In our simulations however, the emerging spatial organization of the synthetic community is a feedback of self-engineered nutrient landscapes that drive localized growth conditions. This highlights the interplay of biotic and abiotic factors during community assembly of sessile bacterial assemblages, and the importance of localized growth rates as a key driver for emerging spatial self-organization.

Genetic segregation of fluorescently labelled strains in experiments suggests a strong reduction in genetic diversity compared to the ancestral population[11]. Limited by the availability of differentiable fluorescent proteins, experimental observations are restricted to a coarse resolution and are not capable of resolving individual lineages, although efforts have been made to include multiple fluorescent proteins[13], high temporal resolution imaging[36], and also three dimensional tracking[37]. An individual-based model on the other hand is capable to resolve the segregation dynamics based on individual inoculated cells, and enabled us to quantify the number of individual cell lineages persisting at the expanding colony edge depending on trophic interaction. Interestingly, comparison of demixing dynamics show a similar trend for strong mutualism and competition when looking at individual lineages compared to the strain-based differentiation, i.e., a higher intermixing in the case of the mutualistic scenario versus competitive (Fig. 3b, c). In the weak mutualistic case, the producer benefits from the reduced competition for space (since the consumer initially does not grow due to the lack of nitrite) resulting in more individual cell lineages proliferating and persisting at the colony periphery. Only once the consumer super-sectors begin to emerge do they begin to pose a barrier for the producer cells resulting in coalescence events and an overall reduction of observed producer cell lineages at the colony periphery. For the case of consumer cells, weak mutualistic growth results in a few "lucky cells" that are able to proliferate and contribute disproportionally to the colony expansion. These cell lineages, despite being dependent on another strain, profit from the weak mutualistic growth, which highlights the somewhat counterintuitive situation of having a higher reproductive success compared to any other scenario despite an initial growth rate disadvantage.

Bacterial assemblages rarely grow on homogeneous surfaces and the spatial context has been shown to strongly influence the community stability and function with respect to trophic interactions[38]. The influence of single objects and rough agar on the proliferation of cell lineages has been observed both experimentally and in silico[39–41]. Specifically, a recent study investigated the growth and spatial self-organization of trophically interacting synthetic communities during range expansion with the inclusion of a spatially structured habitat[42]. Other recent studies include the interrelation of structure and flow environment on biofilm growth[43], the influence of hydration conditions[7] and boundary deformities[42] on the spatial organization of trophically interacting communities[7] and the influence of heterogeneous environments on population genetics[41]. Simplified, cell lineages directly in front of the obstacles are outgrown by lineages close to the obstacle boundary, posing spatial genetic bottlenecks[19,39,40]. When growing in a habitat populated by numerous solid obstacles, this effect is amplified and results in a more pronounced genetic segregation compared to the homogeneous environment void of obstacles with an overall reduction of individual cell lineages residing at the colony periphery. Because natural habitats are rarely homogeneous (as growth on agar pates and in some of the simulations), we explored scenarios with stochastic variations in the size and spacing of obstacles (solid grains or pillars) that showed a similar trend when compared to the simulation results with regular spaced grids (SI Fig. 2, SI Table 1).

Within the simulations, cells were treated as isogenic without any genetic variation or potential for mutations. In natural bacterial assemblages, a higher number of individual cell lineages containing genetic variation and potential mutations at the expanding edge increases the opportunity for natural selection to shape the final community structure[44]. It has been shown previously that allele surfing promotes adaptation of bacterial colonies using a mutation with tunable selective advantage[36]. Again, one of the main differences between this and our system is the fact that the relative growth rate advantage is a dynamic emergent property of the underlying nutrient landscape. Thus, the observed mechanisms of localized growth rates shaping the dynamics of individual cell lineages has the potential of modulating evolutionary processes by determining the standing genetic variation of the expanding edge in natural bacterial assemblages. This in turn accelerates (increased beneficial mutations at the periphery) or decelerates (burying beneficial mutations behind the expanding front) the pace of evolution in bacterial assemblages undergoing range expansion[36]. It is generally thought that a balance between genetic drift and natural selection shapes the population genetics of range expanding communities. We here showed in addition how trophic interactions and nutrient availability underlying bacterial communities can shape the proliferation of individual cells through localized growth rate advantages, and thus alter evolutionary mechanisms acting within the growth layer. Spatially structured habitats mediate the reproductive success of individual lineages and highlight the importance of chance relative to natural selection for trophically interacting bacterial communities. Our simulations and experimental validation emphasize the importance of growth rate differences as a key driver in pattern formation and final genetic population diversity, and the importance of microbes as "environmental engineers" playing a crucial role in not only shaping their imminent surroundings but also having far-reaching consequences for community structure and local ecosystem functioning.

## Methods

**Experimental observations using a synthetic bacterial community.** A previously established synthetic cross-feeding bacterial ecosystem composed of two isogenic mutant strains of the bacterium *Pseudomonas stutzeri* A1501[28,45] was used

as a model community (Fig. 1). Briefly, the producer strain is able to reduce nitrate to nitrite but not further to nitrous oxide. The consumer on the other hand can only reduce nitrite to nitrous oxide but is unable to use nitrate. When grown together with nitrate as the growth-limiting resource, the two strains engage in a nitrite cross-feeding interaction. Nitrite is toxic at pH 6.5 but not at pH 7.5[46–49], thus the cross-feeding interaction can be fine-tuned between weak mutualism and stronger mutualism by adjusting the pH of the growth medium or agar plate. When growing aerobically, the manipulated range of pH does not have any experimentally observable effect on liquid growth capabilities[45]. The two strains also contain an IPTG-inducible green or cyan fluorescent protein-encoding gene[28], which enables us to distinguish and quantify the abundance of each strain when grown together[28,31]. Two complete reducers differentiated by the same IPTG-inducible fluorescent proteins are used to create a scenario where both strains compete for nitrate (competition)[50].

**Experimental range expansion.** The range expansion experiments were performed as described in Goldschmidt[28]. We prepared overnight cultures of the complete reducers alone and the cross-feeding isogenic mutants alone in lysogeny broth (LB) medium for a period of 12 h in a shaking incubator at 37 °C at 220 rpm. When the cultures reached stationary phase, the optical density at 600 nm ($OD_{600}$) was adjusted to one. In order to adjust the OD, the cultures were centrifuged at $3615 \times g$ for 8 min, the supernatants were discarded, and the remaining cells were suspended in 1000 μl of 0.9% (w/v) NaCl solution. We then transferred the cultures into a glove box (Coy Laboratory Products, Grass Lake, MI) containing a nitrogen ($N_2$):hydrogen ($H_2$) (97:3) anaerobic atmosphere. Two strains (two complete reducers or producer and consumer) were mixed at a ratio of 1:1 and 1 μl were then deposited onto the center of anaerobic LB agar plate amended with 1 mM of sodium nitrate ($NaNO_3$) and adjusted to pH 7.5 (weak mutualistic conditions) or 6.5 (strong mutualistic conditions) with 0.5 M NaOH or 30% v/v HCl. Plates were incubated in the anaerobic glove box for a period of 2 weeks at room temperature. The colonies were subsequently imaged using a Leica TCS SP5 II confocal microscope (Etzlar, Germany). The colonies were exposed to aerobic conditions for 1 h preceding image acquisition to allow for maturation of the fluorescent proteins.

**Agent-based mathematical model.** The mathematical model combines numerical pseudo two-dimensional nutrient diffusion with an on-grid individual-based representation of bacterial cells following local growth conditions determined using Monod-type kinetics[7,51,52]. A circular domain of 5 mm radius composed of a hexagonal lattice with side length of 20 microns is used as a backbone for diffusion calculation based on one-dimensional Fickian diffusion between nodes whilst respecting mass balance (consumption and production of the respective nutrients) at each node (Eq. 1):

$$\frac{\partial C_{NO_3}}{\partial t} = D \frac{\partial^2 C_{NO_3}}{\partial x^2} - \frac{\mu_{prod}}{Y_{NO_3}} \frac{\partial m_{prod}}{\partial t}$$
$$\frac{\partial C_{NO_2}}{\partial t} = D \frac{\partial^2 C_{NO_2}}{\partial x^2} - \frac{\mu_{cons}}{Y_{NO_2}} \frac{\partial m_{cons}}{\partial t} + \frac{\mu_P}{Y_{NO_3}} \frac{\partial m_{prod}}{\partial t},$$ (1)

where $C_{NO_3}$ and $C_{NO_2}$ is the concentration of nitrate and nitrite [mol/m³], $t$ is time [s], $D$ the diffusion coefficient [m²/s], $x$ the spatial coordinate along the bond [m], $\mu_{prod}$ and $\mu_{cons}$ the growth rate of producer and consumers [1/s], $Y_{NO3}$ and $Y_{NO2}$ the biomass yield on nitrate and nitrite [kgDW/mol] and $m_{prod}$ and $m_{cons}$ the biomass of producer and consumers [kg]. Details of how the individual components are calculated is shown in Eqs. (2–7) with all parameters given in Table 1. Only nitrate and nitrite are included in the simulation since other nutrients are provided in nonlimiting concentrations. A constant peripheral source for nitrate is included (setting the concentration at the boundary nodes to 1 mM before the numerical diffusion step), whereas nitrite is produced locally solely by bacterial metabolism. Due to the size of each hexagon, bacterial cells are essentially represented as super agents[53], and each grid node is inhabited by a single cell (i.e., strains are mutually exclusive).

Bacterial cells are inoculated at the center (inoculation radius of 2 mm) where each node is attributed randomly with a producer or consumer strain. A random inoculation mass for each superagent is chosen between 10 and 100% of the mass at division.

Bacterial cells consume nutrients depending on the trophic interaction. In the competitive scenario, both strains only consume nitrate where the growth rate for a super-agent $i$ (i.e., node $i$ of the grid) is calculated using Eq. 2:

$$\mu_{cd,i} = \mu_{max} * \frac{C_{NO_3,i}}{C_{NO_3,i} + K_{NO_3}},$$ (2)

where $\mu_{cd,i}$ is the local growth rate of the complete reducer [1/s] at node $i$, $\mu_{max}$ the maximum growth rate [1/s], $K_{NO_3}$ the Monod half-saturation coefficient for nitrate [mM] and $C_{NO3,cd}$ the nitrate concentration [mM] at node $i$. When cross feeding on nitrite (weak mutualism, i.e., no nitrite toxicity), the growth rates of the producer and consumer are described by Eqs. (3) and (4), respectively:

$$\mu_{prod,i} = \mu_{max} * \frac{C_{NO_3,i}}{C_{NO_3,i} + K_{NO_3}},$$ (3)

$$\mu_{cons,i} = \mu_{max} * \frac{C_{NO_2,i}}{C_{NO_2,i} + K_{NO_2}},$$ (4)

where $\mu_{prod,i}$ is the local growth rate of producer cell [1/s] at node $i$, $\mu_{cons,i}$ is the local growth rate of consumer cell [1/s] at node $i$, $K_{NO2}$ the Monod half-saturation coefficient for nitrite [mM], and $C_{NO2,i}$ the nitrite concentration [mM] at node $i$. Biomass dynamics follow a simple exponential model as described by Eq. (5):

$$m_{t+1,i} = m_{t,i} + \mu_i * m_{t,i} * \Delta t$$ (5)

where $m_{t+1,i}$ and $m_{t,i}$ is the biomass [kg] in time step $t$ and $t+1$ at node $i$, respectively, and $\Delta t$ the numerical time step [s]. The total nutrient consumed is related to the gain in biomass at node $i$ following a biomass yield coefficient described by Eq. (6)

$$\vartheta_{n,i} = \frac{\mu_i * m_{t,i}}{Y_i},$$ (6)

where $\vartheta_{n,i}$ is the consumption rate of nutrient $n$ at node $i$ [mol/s] and $Y_i$ the biomass yield coefficient [kgDW/mol]. In the case of nitrate consumption by the producer, an equivalent amount of nitrite is produced following stoichiometry. Nitrite toxicity has been shown to affect the growth yield[54]. Thus, for the mutualistic conditions, the biomass yield coefficient for both strains is calculated using Eq. (7):

$$Y_i = \begin{cases} (Y_{max} - Y_{min}) * \frac{K_{inh}}{K_{inh} + C_{NO_2,i}} + Y_{min}, & pH = 6.5 \\ Y_{max}, & pH = 7.5 \end{cases}$$ (7)

where $Y_{max}$ is the maximum biomass yield [kgDW/mol], $Y_{min}$ the minimum biomass yield [kgDW/mol], and $K_{inh}$ the nitrite inhibition coefficient [mM].

Colony expansion in the mathematical model is a combination of cell division and a simple on-grid cell shoving algorithm. Upon reaching intrinsic critical mass for division, a bacterial cell divides into two where one daughter cell occupies the current location and the second either occupies an adjacent cell (if the location is unoccupied or cell shoving is permitted), or is added to the current location in a layer above (pseudo three-dimensional colony growth). From the location of a dividing cell, the shortest path distance to the colony periphery is calculated. If the distance is sufficiently small (<100 μ, five grid cells) all cells along the shortest path are (mathematically) shoved towards the colony periphery, where the current peripheral cell is assigned a new node at random from any of the unoccupied neighboring nodes. The physical obstacles (in the structured scenarios) are treated as solid and impenetrable domain boundaries. The structured habitats were simulated by adding solid obstacles of defined dimensions (100 μ diameter) spaced regularly within the domain (500, 400, and 300 μ for the coarse, medium, and fine structured scenarios). Stochastic obstacle simulations in SI Fig. 2 contained the medium and fine spacing, with obstacle diameters taken from a normal distribution with mean 100 μ, and variance 20 μ. The total simulated time is 72 h using a 60 s time step.

**Table 1 List of modelling parameters and references.**

| Parameter | Abbreviation | Value | Units | Reference |
|---|---|---|---|---|
| Nitrate diffusion coefficient | $D_{NO3}$ | 1.7e−9 | m²s⁻¹ | 56 |
| Nitrite diffusion coefficient | $D_{NO2}$ | 1.7e−9 | m²s⁻¹ | Equal nitrate |
| Maximum growth rate | $\mu_{max}$ | 5.4e−6 | s⁻¹ | 57 |
| Monod nitrate limitation | $K_{lim,NO3}$ | 0.04 | mM | 58 |
| Monod nitrite limitation | $K_{lim,NO2}$ | 0.04 | mM | Equal nitrate |
| Monod nitrite inhibition | $K_{inh,NO2}$ | 0.02 | mM | 59 |
| Maximum yield | $Y_{max}$ | 0.06 | kgDM/mol | 60 |
| Minimum biomass yield | $Y_{min}$ | 0.006 | kgDM/mol | 10% of $Y_{max}$ |

Nitrite diffusion and nitrite limitation were chosen equal to nitrate in order to restrict any bias due to parameterization.

**Statistics and reproducibility**. Following the procedure outlined in the "Methods" section, distinct colony patterns emerged reproducibly for the different trophic interactions. Simulation results were compared to multiple experimental replicates (a single experimental replicate is provided in this study). For the mathematical simulations, six replicates of the nonstructured scenario and ten for all structured scenarios were generated per trophic interaction. Here, a replicate is defined as simulations containing the same initial and boundary conditions, where stochasticity is included through random position during inoculation and during the simulation via the shoving algorithm. When statistically comparing different simulation results (e.g., biomass, number of branches etc.), two-sample $t$-tests were used.

## Data availability

The experimental data (microscopy images) that support the findings of this study are available within the manuscript and supporting information.

## Code availability

The mathematical code that support the findings of this study is available in the ETH Research Collection with the identifier 10.3929/ethz-b-000432690[55].

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

## Acknowledgements

Financial support for this work came from an Advanced Grant to D.O. by the European Research Council (ERC-3200499-'SoilLife') and from the RTD SystemsX.ch project 'MicroscapesX' and grants from the Swiss National Science Foundation (grant numbers 31003A_149304 and 31003A_176101) awarded to D.J.

## Author contributions

B.B., D.C, D.J., and D.O. conceived the study and wrote the manuscript. D.C. performed the experiments including visualization. B.B. wrote the mathematical model, created simulation scenarios and analyzed the data.

## Competing interests

The authors declare no competing interests.
