## [Peer Review File · Communications Biology]

Reviewers' comments:

Reviewer #1 (Remarks to the Author):

This manuscript presents simulations and experiments investigating the spatial structure and interactions between microbes impact range expansions. Three scenarios - competition, weak mutualism, and strong mutualism - are explored.

I found this manuscript interesting; the experiments and simulations appear well-done. Having said that, I have a few questions and comments.

Major issues:

1. I appreciate the inclusion of the shoving algorithm. Indeed, many of the papers cited by the authors demonstrate its importance. However, its role in the current study seems under-explored. For example, on line 134 the authors state "consumer cells are reliant on proliferation and shoving..." but do not actually demonstrate that shoving is responsible. While this seems likely to be true, it is not shown. Certainly starting with zero growth rate is a large disadvantage, but if growth rate increases quickly enough, shoving is not necessary.
2. In line 174-176+ the authors state "Super sectors that penetrate the layer of producers are typically monoclonal, owing to the fact that they typically nucleate from a single cell" This seems like an important point, making it worth quantifying "typically."
3. While I enjoyed the section on structured environments, it made me curious how the regular structure and size of these ordered arrays of 'holes' could impact the outcome. As 'lucky' events are important, could homogeneity suppress them more than heterogeneous environments?
4. I found figure 4 difficult to interpret. It appears the plots on the bottom row are associated with A, B, and C? This would be clearer if all panels were labeled A-F. Further, I am interested in seeing all quantities measured for all scenarios.

Minor comments:

1. It's a small detail, but I did not see the size or spacing of the 'holes' for the structured environment reported. This information would be important for someone to replicate the simulations.
2. Experiments testing the structured environment simulations would be very interesting if possible, but are not required for publication, in my opinion.
3. It may be worth citing Lowery, Vallespir and Ursell "Structured environments fundamentally alter dynamics and stability of ecological communities." PNAS (2019). While this paper does not investigate range expansions, it does study the effect of spatial structure on competition.
4. This is purely the authors' choice, but the title primarily refers to one of the three scenarios investigated here. While it is, to me, the most interesting of the three, I was surprised that the paper was more balanced between the three cases than the title might suggest.

Reviewer #2 (Remarks to the Author):

In this manuscript, the authors used agent-based mathematical model to track individual cell lineages and more interestingly explored the range expansion interrupted by physical obstacles. Combining experiments and model to shape the community patterns during bacterial range expansion is of importance and not a trivial effort. Also, attempt to identify the lineage of community pattern is worthwhile. However, I am not convinced of the value of the model, as

explained below.

1. The insight gained in this study was based on a 2-dimensional model. However, the colony growth is a typically 3-dimensional structure formation process. Bacterial cells grow/expand vertically and horizontally. The distributions of bacterial cells and nutrient in the colony at Z-axis have been recently discussed. e.g., eLife 8: e41093 (2019). The main results including lineage tracking and habitat structure might be very different if a 3D model was applied.

2. In the agent based model, the authors used a simple ongrid shoving algorithm to model the colony expansion. However, the actual shoving process is a stochastic one. How would this affect the lineage tracking? Besides, in the presence of obstacles, the authors didn't clearly explain how they handled the boundary conditions where the cells crash to the obstacles. I would expect the types of boundary conditions can result different cell lineage structures in space.

Minor points:

I found that figure 2ab is rather difficult to understand.

Reviewer #3 (Remarks to the Author):

In manuscript COMMSBIO-20-0997-T the range expansion of surface-attached cross-feeding bacterial communities is studied under different interaction scenarios, going from competition for two complete reducer strains to weak or strong mutualism for two complementary strains, a producer that reduces nitrates to nitrite, and a consumer that uses nitrite. The level of mutualism is controlled by the pH of the medium in the experiments, which controls the toxicity of the nitrite. Low toxicity for high pH implies weak mutualism, while high toxicity for low pH induces strong mutualism as the producer can not live without the consumer. This interaction, mediated by the diffusion of nitrate and nitrite in space induces self-organization in space, displaying different patterns in each case. They study also the effect of a spatial structure in the expansion of the colony. They find that structured space reduce in general the number of cell linages because some get blocked at the obstacles.

This work reports a detailed analysis including experimental and numerical results. The results are interesting and convincing, and shed light on the role of the interaction and spatial inhomogeneities in shaping bacterial communities. The article is also clearly written and reasonable easy to follow by a broad audience. There are however a few point (some minor) that need clarification before I can recommend the manuscript for publication in Nature Communications Biology.

1. In my opinion the title of this work, if catchy, is not informative enough. I would say that the fact that some linages proliferate by change is not the main result of the paper, but the identification of the patterns emerging for each different type of interaction and spatial structure is. I would suggest the title to reflect the more general results and not focus only on the "lucky individuals".

2. At some point is not clear to me what is new and what is not with respect to previous literature. Is the model new? Is it the technique to control the interaction type with the pH? The introduction of spatial structures? This should be clearly explained in the text.

3. Only periodic arrays of obstacles are considered. How would the results change if the obstacles were random in size as well as their spatial distribution? By the way, how the spatial inhomogeneities are implemented in the model is not explained.

4. Concerning the model, it is not explained how the consumption of nutrients, R in Eq. (1), depends on the biomass. I understand it through Eq. (6) but it could be explicitly written. The

creation of nitrite is mentioned after Eq. (6), but I would prefer to see the set of coupled differential equations for the concentrations of nitrate, nitrite and biomass of each strain explicitly written. Also how the source of nitrate is included in the equations should be mentioned.

5. The authors consider one dimensional diffusion between nodes on a hexagonal lattice. This is a little strange to me. I would find more natural to implement a two dimensional diffusion on the hexagonal lattice. This is not a minor difference, as considering the one dimensional diffusion between nodes does not properly implement the role of the curvature of concentration fronts in the dynamics.

6. The rule for cell division is not explained either.

7. There is a typo in line 133.

Editor's comments

As pointed out by Referee #1 and #2, several other models and references on this topic are neglected here. For example, how can you distinguish rare successful positioning with noise fluctuation into self-organized patterns within microbial systems? Do these patterns represent phase separation or Turing patterns? How do these processes relate to your work? You may discuss them within the discussion section.

We would like to thank the editor for posing these fundamental questions.

- 1) *How can you distinguish rare successful positioning with noise fluctuation into self-organized patterns within microbial systems?*

In the simulations, rare successful positioning is a stochastic result of the initial spatial inoculation of the cells and is thus interrelated to noise fluctuations. If however the initial spatial positioning is favourable, localized conditions facilitate the emergence of a super sector with the exception of coalescence events burying the sector in early stages of the colony development. Thus, noise fluctuations in our simulations are included in the random spatial inoculation of the colony, which determines the favourable initial spatial positioning for consumer super sectors to emerge.

- 2) *Do these patterns represent phase separation or Turing patterns? How do these processes relate to your work?*

Indeed the producer and consumer may be viewed as classical inhibitors, activators or a mixture of the two (producer in strong mutualistic conditions) in the classical Turing theory. Similarly, there is a resemblance of density dependant motility through the shoving mechanism as in the theory of phase separation. However, the main difference separating our observations from the above mentioned theories is the fact that the diffusion fluxes of nitrate and nitrite as well as the actual bacterial motility is a localized phenomena in contrast to the system properties as assumed in both the Turing pattern processes or phase separation. Moreover, the actors are not uniformly distributed in space (which is a central point in this work) and thus, in addition to the diffusion-reaction waves favoured lineages amplify or suppress such waves based on their trophic preferences.

Therefore the patterns emerging in our simulations cannot be compared to either of the above mentioned theories but represents a case where localized conditions, self-engineered metabolic landscapes and initial spatial positioning govern the final spatial organization of the community.

We thank the reviewer for pointing this out and we have added a paragraph in the introduction to specify the difference in our observed pattern formation mechanisms to the Turing pattern formation or phase separation mechanisms.

Lines 293 - 297: "In comparison to other processes that result in spatially self-organized patterns such as the classical Turing pattern processes¹ or phase separation², the spatial organization in our model is governed by localized growth conditions, self-engineered metabolic landscapes and favourable initial spatial positioning that differ fundamentally from the system properties in both the above mentioned theories"

General comments by the authors.

First, we thank the reviewers for their insightful and constructive comments.

Overall, we changed the title of the manuscript to better represent all three cases of trophic interactions and their specific spatial patterns induced by self-engineered metabolic landscapes. We improved the materials and methods section by including additional descriptions concerning nutrient diffusion equations, the cell division calculation and details on spacing etc. of the structured habitat simulations. We added additional simulation results to provide evidence for the importance of shoving for the spatial self-organization (SI Figure 1) and how structured environments with stochastic arrangement influences the spatial self-organization (SI Figure 2).

In the following, we provide a point-by-point response to reviewers' comments. All indication of lines are taken from the cleaned version of the manuscript (i.e. not track-changed document).

Reviewer #1 (Remarks to the Author):

This manuscript presents simulations and experiments investigating the spatial structure and interactions between microbes impact rang expansions. Three scenarios - competition, weak mutualism, and strong mutualism - are explored.

I found this manuscript interesting; the experiments and simulations appear well-done. Having said that, I have a few questions and comments.

We would like to thank the reviewer for the time taken to read, comment and improve the manuscript in the process. Especially, we created two supplementary information figures concerning the relevance of the shoving algorithm and stochastic obstacles in the structured environment scenarios. Below are additional answers to all concerns raised by the reviewer.

Major issues:

1. I appreciate the inclusion of the shoving algorithm. Indeed, many of the papers cited by the authors demonstrate its importance. However, its role in the current study seems under-explored. For example, on line 134 the authors state "consumer cells are reliant on proliferation and shoving..." but do not actually demonstrate that shoving is responsible. While this seems likely to be true, it is not shown. Certainly starting with zero growth rate is a large disadvantage, but if growth rate increases quickly enough, shoving is not necessary.

Indeed this is an important omission, especially since one of the main mechanisms for consumer branches emerging in weak mutualistic conditions is through initial shoving by the producers and subsequent proliferation. To highlight this, we created additional simulations which include the omission of the shoving algorithm (i.e. dividing cells can only place their daughter cells in adjacent unoccupied grid nodes) and by showing the effect when a certain cell type occupies the whole colony periphery during inoculation (but with the inclusion of cell shoving). The results are shown in the Figure below. Omitting cell shoving results in a lineage and sector development perpendicular to the colony periphery (Panels a, b and c for competition, weak mutualism and strong mutualism respectively). In case of weak mutualism, a reduced number of consumer lineages is visible due to their initial growth deficit lacking nitrite. Any branch that does emerge comes from a cluster of cells that is not covered by adjacent producers during the initial growth phase. When artificially coating

the whole colony with consumer cells (Panel d) the resulting colony is dominated by consumers at the periphery due to the large number of consumer branches emerging that benefit from their initial spatial positioning. When the whole colony is artificially coated with producer cells, no consumer branches are observed since the mechanism of being initially shoved by producer cells is not possible. These additional simulations confirm our hypothesis of how the consumer branches emerge in silico under weak mutualistic conditions.

We added a description of the importance of cell shoving and initial spatial positioning to the manuscript.

Lines 276 - 287: "An in depth experimental evaluation of the importance of initial spatial positioning and the role of cell shoving is complex due to massive amount of replicates and inability to remove cell shoving in a living community. Alternatively we have removed the shoving component in the mathematical model (i.e. allowing only peripheral cells contribution to colony expansion through placement of their progeny on unoccupied grid nodes). The result was an overall lower expansion of the colonies (SI Figure 1a,b and c) with conspicuous omission of super sectors in the weak mutualistic scenario (SI Figure 1b). Additionally, we could easily control the initial positioning of consumers and producers in the mathematical model. Encasing the colony with consumer cells resulted in myriads of consumer super sectors (SI Figure 1d) whereas encasing the inoculated colony with producer cells suppressed the emergence of any consumer super sectors as they are buried behind the advancing producer front. These additional simulations highlight the importance of initial spatial positioning and cell shoving for the seeding of the observed consumer super sectors in case of the weak mutualistic scenario."

SI Figure 1: Importance of bacterial shoving and initial spatial positioning for pattern formation. Removal of the shoving algorithm (i.e. only cells at the colony periphery can contribute to colony expansion through placing their progeny on unoccupied grid nodes) results in a similar pattern (although reduce in expansion) for the competitive scenario (a), a suppression of super sectors in case of weak mutualism (b) and a significantly reduced expansion rate in the strong mutualistic scenario (c). In case of weak mutualistic conditions, initial positioning of consumer cells is fundamental for colony pattern formation. When the whole periphery is occupied by consumer cells (d), numerous consumer super branches emerge since consumer cells are frequently pushed ahead by the proliferating producer. When the inoculated colony periphery is dominated by producer cells (e), no consumer branches emerge since the consumer is buried behind the advancing producers.

2. In line 174-176+ the authors state "Super sectors that penetrate the layer of producers are typically monoclonal, owing to the fact that they typically nucleate from a single cell" This seems like an important point, making it worth quantifying "typically."

We have quantified the average number of lineages per branch in the simulations and added these data to the manuscript.

Lines 176 - 178: "Super sectors that penetrate the layer of producers are typically monoclonal, owing to the fact that they typically nucleate from a single cell (ratio of consumer lineages at the periphery to consumer branches is 1.25 ± 0.05)."

3. While I enjoyed the section on structured environments, it made me curious how the regular structure and size of these ordered arrays of 'holes' could impact the outcome. As 'lucky' events are important, could homogeneity suppress them more than heterogeneous environments?

This is indeed an interesting point and we have expanded the geometries in the model to include two stochastic domains that vary in pillar radii and density. The results have been added as a visualization to the supplementary material (SI Figure 2) and is shown below for simplicity. The addition of stochastic pillar size results in a heterogeneous pore space similar to the case where the spatial positioning of homogeneous pillars would be chosen stochastically and we thus only created simulations with stochastic pillar size with a homogeneous spacing.

Overall, we find that a structured environment with stochastic pillars created more bottlenecks for bacterial expansion, which results in more coalescence events and less lineages persisting to the colony periphery. In short, the addition of stochastic pillars amplified the findings of the analysis when only using regular grids as a structured environment.

We referred to these simulations in the manuscript in the discussion section.

Lines 344 - 347: "Because natural habitats are rarely homogeneous (as growth on agar plates and in some of the simulations), we explored scenarios with stochastic variations in the size and spacing of obstacles (solid grains or pillars) that showed a similar trend when compared to the simulation results with regular spaced grids (SI Figure 2, SI Table 1)."

SI Figure 2: Inclusion of stochastic variations in particle size simulations of structured habitats with three different obstacle scaping. The inclusion of additional heterogeneity through stochastic variation of obstacle diameter whilst keeping the spacing equal results in an amplified effect of lineage coalescence for all scenarios and a reduction in observed consumer branches in the weak and strong mutualistic scenarios.

SI Table 1: Quantification of total number of peripheral lineages [#], persistent consumer branches [#] and biomass [10^{-8} kg] for competition (CT), weak mutualism (WM) and strong mutualism (SM) in all simulated habitat scenarios. Table shows mean values of all replicates including standard deviations in brackets.

	Total lineages [#]			Consumer branches [#]			Biomass [kg]		
	CT	WM	SM	CT	WM	SM	CT	WM	SM
Homogeneous	163 (13)	143 (12)	283 (12)	85 (7)	22 (6)	110 (5)	3.43 (0.05)	3.18 (0.10)	1.74 (0.01)
Regular coarse	107 (3)	89 (7)	173 (4)	52 (2)	27 (6)	68 (5)	2.96 (0.02)	2.70 (0.06)	1.56 (0.01)
Regular medium	87 (6)	73 (6)	134 (18)	44 (6)	19 (3)	51 (8)	2.56 (0.24)	2.31 (0.22)	1.37 (0.02)
Regular fine	68 (4)	65 (5)	100 (5)	33 (6)	14 (4)	36 (3)	1.91 (0.03)	1.71 (0.03)	1.03 (0.01)
Stochastic coarse	100 (5)	86 (8)	175 (8)	49 (4)	25 (8)	66 (4)	2.95 (0.03)	2.69 (0.06)	1.55 (0.01)
Stochastic medium	85 (3)	73 (6)	143 (7)	40 (5)	23 (6)	54 (4)	2.64 (0.03)	2.37 (0.03)	1.36 (0.01)
Stochastic fine	62 (4)	56 (3)	98 (3)	30 (5)	15 (2)	38 (5)	1.85 (0.05)	1.62 (0.04)	1.00 (0.02)

4. I found figure 4 difficult to interpret. It appears the plots on the bottom row are associated with A, B, and C? This would be clearer if all panels were labeled A-F. Further, I am interested in seeing all quantities measured for all scenarios.

We have relabelled the panels from a) through to f) for clarification and updated the Figure legend and manuscript text where necessary. In addition, we added a table to the supplementary information that includes all the quantifications for all scenarios including the stochastic particle size variation.

Minor comments:

1. It's a small detail, but I did not see the size or spacing of the 'holes' for the structured environment reported. This information would be important for someone to replicate the simulations.

As it is an important omission we have added the description to the methods section for all scenarios.

Lines 451 - 456: "The physical obstacles (in the structured scenarios) are treated as solid and impenetrable domain boundaries. The structured habitats were simulated by adding solid obstacles of defined dimensions (100 microns diameter) spaced regularly within the domain (500 microns, 400 microns and 300 microns for the coarse, medium and fine structured scenarios). Stochastic obstacle simulations in SI Figure 2 contained the medium and fine spacing, with obstacle diameters taken from a normal distribution with mean 100 microns and variance 20 microns."

2. Experiments testing the structured environment simulations would be very interesting if possible, but are not required for publication, in my opinion.

Indeed, we would ourselves be interested in such experiments. In a similar experimental setup when compared to our simulation domains, Ciccarese et al. have performed experiments in structured environments to observe the effect of spatial structure on self-organization dynamics³. Experiments that have controllable boundary conditions similar to the mathematical model would be intricate to perform (the actual physical domain, boundary conditions and visualization). We have included a more prominent reference to the above-mentioned paper in the manuscript.

Lines 334 - 336: "Specifically, a recent study investigated the growth and spatial self-organization of trophically interacting synthetic communities during range expansion with the inclusion of a spatially structured habitat³."

3. It may be worth citing Lowery, Vallespir and Ursell "Structured environments fundamentally alter dynamics and stability of ecological communities." PNAS (2019). While this paper does not investigate range expansions, it does study the effect of spatial structure on competition.

We thank the reviewer for pointing towards this publication.

Lines 331 - 332: "Bacterial assemblages rarely grow on homogeneous surfaces and the spatial context has been shown to profoundly influence the community stability and function with respect to trophic interactions⁴."

4. This is purely the authors' choice, but the title primarily refers to one of the three scenarios investigated here. While it is, to me, the most interesting of the three, I was surprised that the paper was more balanced between the three cases than the title might suggest.

To better represent all trophic interactions and resulting spatial organizations in response to metabolic landscape feedbacks we have changed the title to:

"Spatial organization in microbial range expansion emerging from trophic dependencies and successful lineages"

Reviewer #2 (Remarks to the Author):

In this manuscript, the authors used agent-based mathematical model to tracks individual cell lineages and more interestingly explored the range expansion interrupted by physical obstacles. Combining experiments and model to shape the community patterns during bacterial range expansion is of importance and not a trivial effort. Also, attempt to identify the lineage of community pattern is worthwhile. However, I am not convinced of the value of the model, as explained below.

We would like to thank reviewer 2 for the constructive comment. We included additional information concerning the 3D structure of colonies to the manuscript and how boundary conditions are handled in the structured habitat scenarios.

1. The insight gained in this study was based on a 2-dimensional model. However, the colony growth is a typically 3-dimensional structure formation process. Bacterial cells grow/expand vertically and horizontally. The distributions of bacterial cells and nutrient in the colony at Z-axis have been recently discussed. e.g., eLife 8: e41093 (2019). The main results including lineage tracking and habitat structure might be very different if a 3D model was applied.

Indeed, the dimensionality question raised by the reviewer is a very important one and there is no simple experimental method for testing the outcomes for the colony size in this manuscript. Single cell resolution tracking and the visualization of three-dimensional structures in bacterial micro colonies has been achieved recently⁵, however the tracking capabilities do not extend to colonies the size of our experimental system. Furthermore, the primary region of growth and nutrient penetration is restricted to base of the colony (approximately 10 microns)⁶, which is also the part of the colony that primarily influences the underlying nutrient landscape⁷. Our mathematical model does not resolve the physics of bacterial shoving at the single cell level due to the representation of bacterial cells as super-agents. Although incorporating a three-dimensional shoving algorithm may give further insights into the internal structure of the colony, these results may be difficult to validate due to the coarse representation of three-dimensional shoving and lack of experimental data and thus remain speculative.

2. In the agent based model, the authors used a simple ongrid shoving algorithm to model the colony expansion. However, the actual shoving process is a stochastic one. How would this affect the lineage tracking? Besides, in the presence of obstacles, the authors didn't clearly explain how they handled the boundary conditions where the cells crash to the obstacles. I would expect the types of boundary conditions can result different cell lineage structures in space.

Overall, the shoving algorithm includes two stochastic elements. Firstly, although the shortest path between the cell and the colony periphery is used for shoving, there are often more than one path along the grid, which are of the same shortest length. Under those conditions, a shortest path is chosen at random between all variants. Secondly, the cell residing at the colony periphery is allocated an adjacent, unoccupied grid node at random which may alter the proliferation of different lineages.

In the event that a cell collides with the domain boundary or a pillar in the structured habitat, we added clarification in the materials and methods section.

Lines 451 - 452: "The physical obstacles (in the structured scenarios) are treated as solid and impenetrable domain boundaries."

Minor points:

I found that figure 2ab is rather difficult to understand.

For clarification, we split nitrate and nitrite in panel 2b and added further explanation to the Figure caption.

Reviewer #3 (Remarks to the Author):

In manuscript COMMSBIO-20-0997-T the range expansion of surface-attached cross-feeding bacterial communities is studied under different interaction scenarios, going from competition for two complete reducer strains to weak or strong mutualism for two complementary strains, a producer that reduces nitrates to nitrite, and a consumer that uses nitrite. The level of mutualism is controlled by the pH of the medium in the experiments, which controls the toxicity of the nitrite. Low toxicity for high pH implies weak mutualism, while high toxicity for low pH induces strong mutualism as the producer can not live without the consumer. This interaction, mediated by the diffusion of nitrate and nitrite in space induces self-organization in space, displaying different patterns in each case. They study also the effect of a spatial structure in the expansion of the colony. They find that structured space reduce in general the number of cell lineages because some get blocked at the obstacles.

This work reports a detailed analysis including experimental and numerical results. The results are interesting and convincing, and shade light on the role of the interaction and spatial inhomogeneities in shaping bacterial communities. The article is also clearly written and reasonable easy to follow by a broad audience. There are however a few point (some minor) that need clarification before I can recommend the manuscript for publication in Nature Communications Biology.

We would like to thank the reviewer for the thoughtful and constructive comments. Since some issues raised are similar to those of reviewer 1, we have referred to those descriptions further above.

1. In my opinion the title of this work, if catchy, is not informative enough. I would say that the fact that some lineages proliferate by change is not the main result of the paper, but the identification of the patterns emerging for each different type of interaction and spatial structure is. I would suggest the title to reflect the more general results and not focus only on the "lucky individuals".

In order to better represent all trophic interactions and resulting spatial organizations in response to metabolic landscape feedbacks we have changed the title to:

"Spatial organization in microbial range expansion emerging from trophic dependencies and successful lineages"

2. At some point is not clear to me what is new and what is not with respect to previous literature. Is the model new? Is it the technique to control the interaction type with the pH? The introduction of spatial structures? This should be clearly explained in the text.

We have highlighted more prominently that the main novelty of this manuscript is the mechanism of feedbacks between the emerging colony spatial organization and self-engineered nutrient landscapes depending on trophic interactions.

Lines 66 - 69: "Our modelling framework provides novel mechanistic insights into experimental observations of spatial self-organization during bacterial range expansion, predicting a feedback between the emerging colony pattern and self-engineered nutrient landscape at a resolution currently inaccessible to experimental quantification."

Lines 302 - 306: "Although the resulting colony pattern in our system is similar to the above-mentioned studies, where growth rate differences are imposed via mutations and are thus fixed in time, the mechanism is fundamentally different. In our simulations however, the emerging spatial organization of the synthetic community is a feedback of self-engineered nutrient landscapes that drive localized growth conditions."

We also clarified that the experimental setup and method of tuning trophic interactions has been established previously and is not a novelty of this study.

*Lines 373 - 374: "A previously established synthetic cross-feeding bacterial ecosystem composed of two isogenic mutant strains of the bacterium *Pseudomonas stutzeri* A1501^{8,9} was used as a model community (Fig. 1)."*

3. Only periodic arrays of obstacles are considered. How would the results change if the obstacles were random in size as well as their spatial distribution? By the way, how the spatial inhomogeneities are implemented in the model is not explained.

We have added simulations with stochastic distributions of the different pillar geometries as supplementary information. Details of our findings are outlined in the answer to reviewer 1 major comment 3 above.

4. Concerning the model, it is not explained how the consumption of nutrients, R in Eq. (1), depends on the biomass. I understand it through Eq. (6) but it could be explicitly written. The creation of nitrite is mentioned after Eq. (6), but I would prefer to see the set of coupled differential equations for the concentrations of nitrate, nitrite and biomass of each strain explicitly written. Also how the source of nitrate is included in the equations should be mentioned.

We have expanded Equation 1 to include a description of how nitrate and nitrite are consumed and produced by the two strains in the form of a coupled system of differential equations. Since the strains obey different kinetic laws depending on the trophic interaction, listing equations for all strains is impractical and would result in a large table of differential equations. We have added information on the nitrate source for the simulations.

Lines 414 - 417: "A constant peripheral source for nitrate is included (setting the concentration at the boundary nodes to 1 mM before the numerical diffusion step) whereas nitrite is produced locally solely by bacterial metabolism."

5. The authors consider one dimensional diffusion between nodes on a hexagonal lattice. This is a little strange to me. I would find more natural to implement a two dimensional diffusion on the hexagonal lattice. This is not a minor difference, as considering the one dimensional diffusion between nodes does not properly implement the role of the curvature of concentration fronts in the dynamics.

We agree with the reviewer that there is an inherent bias when using pseudo two-dimensional diffusion represented as one-dimensional diffusion between nodes as shown in the Figure below (comparing our algorithm of 1D diffusion on a hexagonal grid with 2D diffusion on a rectangular grid). In this example, diffusion of an arbitrary concentration is calculated for an arbitrary space with unified time (30 time steps). Although there is some slight deviation between the two, the overall spatial distribution and magnitude is very similar. The main advocate for using a hexagonal grid lies in its simultaneous use for the shoving algorithm. Compared to a square grid, a hexagonal grid provides a more natural geometry for the shoving algorithm. Evidently, a two-dimensional diffusion algorithm could be coupled with a hexagonal shoving grid (or off-grid shoving algorithm), however due to the increase in computational complexity and potential numerical inaccuracies (interpolating the concentration experienced by each cell on the hexagonal grid from a rectangular grid) we

concluded that the one-dimensional diffusion algorithm is adequate for this model.

6. The rule for cell division is not explained either.

We have added the omitted information concerning cell division to the manuscript.

Lines 444 - 447: " Upon reaching intrinsic critical mass for division, a bacterial cell divides into two where one daughter cell occupies the current location and the second either occupies an adjacent cell (if the location is unoccupied or cell shoving is permitted) or is added to the current location in a layer above (pseudo three-dimensional colony growth)."

7. There is a typo in line 133.

We have corrected the typo in the manuscript – many thanks.

References:

1. Karig, D. et al. *Stochastic Turing patterns in a synthetic bacterial population. Proc. Natl. Acad. Sci. U. S. A. (2018).*
2. Liu, Q. X. et al. *Phase separation explains a new class of selforganized spatial patterns in ecological systems. Proc. Natl. Acad. Sci. U. S. A. (2013).*
3. Ciccarese, D., Zuidema, A., Merlo, V. & Johnson, D. R. *Interaction-dependent effects of surface structure on microbial spatial self-organization. Philos. Trans. R. Soc. Lond. B. Biol. Sci. (2020).*
4. Lowery, N. V. & Ursell, T. *Structured environments fundamentally alter dynamics and stability of ecological communities. Proc. Natl. Acad. Sci. U. S. A. 16, 379–388 (2019).*
5. Hartmann, R. et al. *Emergence of three-dimensional order and structure in growing biofilms. Nature Physics (2019).*
6. Warren, M. R. et al. *Spatiotemporal establishment of dense bacterial colonies growing on hard agar. Elife (2019).*
7. Cole, J. A., Kohler, L., Hedhli, J. & Luthey-Schulten, Z. *Spatially-resolved metabolic cooperativity within dense bacterial colonies. BMC Syst. Biol. 9, 1–17 (2015).*
8. Lilja, E. E. & Johnson, D. R. *Segregating metabolic processes into different microbial cells accelerates the consumption of inhibitory substrates. ISME J. 10, 1568–1578 (2016).*
9. Goldschmidt, F., Regoes, R. R. & Johnson, D. R. *Successive range expansion promotes diversity and accelerates evolution in spatially structured microbial populations. ISME J. (2017).*

REVIEWERS' COMMENTS:

Reviewer #1 (Remarks to the Author):

I thank the authors for their thorough and interesting responses to my questions and comments, as well as those from the editor and other referees. This is an interesting and timely paper; I happily support publication.

Reviewer #2 (Remarks to the Author):

The authors have addressed all my concerns. I am happy with this revision and I think that the manuscript in its present form should be accepted.

Editor's comments

As pointed out by Referee #1 and #2, several other models and references on this topic are neglected here. For example, how can you distinguish rare successful positioning with noise fluctuation into self-organized patterns within microbial systems? Do these patterns represent phase separation or Turing patterns? How do these processes relate to your work? You may discuss them within the discussion section.

We would like to thank the editor for posing these fundamental questions.

- 1) *How can you distinguish rare successful positioning with noise fluctuation into self-organized patterns within microbial systems?*

In the simulations, rare successful positioning is a stochastic result of the initial spatial inoculation of the cells and is thus interrelated to noise fluctuations. If however the initial spatial positioning is favourable, localized conditions facilitate the emergence of a super sector with the exception of coalescence events burying the sector in early stages of the colony development. Thus, noise fluctuations in our simulations are included in the random spatial inoculation of the colony, which determines the favourable initial spatial positioning for consumer super sectors to emerge.

- 2) *Do these patterns represent phase separation or Turing patterns? How do these processes relate to your work?*

Indeed the producer and consumer may be viewed as classical inhibitors, activators or a mixture of the two (producer in strong mutualistic conditions) in the classical Turing theory. Similarly, there is a resemblance of density dependant motility through the shoving mechanism as in the theory of phase separation. However, the main difference separating our observations from the above mentioned theories is the fact that the diffusion fluxes of nitrate and nitrite as well as the actual bacterial motility is a localized phenomena in contrast to the system properties as assumed in both the Turing pattern processes or phase separation. Moreover, the actors are not uniformly distributed in space (which is a central point in this work) and thus, in addition to the diffusion-reaction waves favoured lineages amplify or suppress such waves based on their trophic preferences.

Therefore the patterns emerging in our simulations cannot be compared to either of the above mentioned theories but represents a case where localized conditions, self-engineered metabolic landscapes and initial spatial positioning govern the final spatial organization of the community.

We thank the reviewer for pointing this out and we have added a paragraph in the introduction to specify the difference in our observed pattern formation mechanisms to the Turing pattern formation or phase separation mechanisms.

Lines 293 - 297: "In comparison to other processes that result in spatially self-organized patterns such as the classical Turing pattern processes¹ or phase separation², the spatial organization in our model is governed by localized growth conditions, self-engineered metabolic landscapes and favourable initial spatial positioning that differ fundamentally from the system properties in both the above mentioned theories"

General comments by the authors.

First, we thank the reviewers for their insightful and constructive comments.

Overall, we changed the title of the manuscript to better represent all three cases of trophic interactions and their specific spatial patterns induced by self-engineered metabolic landscapes. We improved the materials and methods section by including additional descriptions concerning nutrient diffusion equations, the cell division calculation and details on spacing etc. of the structured habitat simulations. We added additional simulation results to provide evidence for the importance of shoving for the spatial self-organization (SI Figure 1) and how structured environments with stochastic arrangement influences the spatial self-organization (SI Figure 2).

In the following, we provide a point-by-point response to reviewers' comments. All indication of lines are taken from the cleaned version of the manuscript (i.e. not track-changed document).

Reviewer #1 (Remarks to the Author):

This manuscript presents simulations and experiments investigating the spatial structure and interactions between microbes impact rang expansions. Three scenarios - competition, weak mutualism, and strong mutualism - are explored.

I found this manuscript interesting; the experiments and simulations appear well-done. Having said that, I have a few questions and comments.

We would like to thank the reviewer for the time taken to read, comment and improve the manuscript in the process. Especially, we created two supplementary information figures concerning the relevance of the shoving algorithm and stochastic obstacles in the structured environment scenarios. Below are additional answers to all concerns raised by the reviewer.

Major issues:

1. I appreciate the inclusion of the shoving algorithm. Indeed, many of the papers cited by the authors demonstrate its importance. However, its role in the current study seems under-explored. For example, on line 134 the authors state "consumer cells are reliant on proliferation and shoving..." but do not actually demonstrate that shoving is responsible. While this seems likely to be true, it is not shown. Certainly starting with zero growth rate is a large disadvantage, but if growth rate increases quickly enough, shoving is not necessary.

Indeed this is an important omission, especially since one of the main mechanisms for consumer branches emerging in weak mutualistic conditions is through initial shoving by the producers and subsequent proliferation. To highlight this, we created additional simulations which include the omission of the shoving algorithm (i.e. dividing cells can only place their daughter cells in adjacent unoccupied grid nodes) and by showing the effect when a certain cell type occupies the whole colony periphery during inoculation (but with the inclusion of cell shoving). The results are shown in the Figure below. Omitting cell shoving results in a lineage and sector development perpendicular to the colony periphery (Panels a, b and c for competition, weak mutualism and strong mutualism respectively). In case of weak mutualism, a reduced number of consumer lineages is visible due to their initial growth deficit lacking nitrite. Any branch that does emerge comes from a cluster of cells that is not covered by adjacent producers during the initial growth phase. When artificially coating

the whole colony with consumer cells (Panel d) the resulting colony is dominated by consumers at the periphery due to the large number of consumer branches emerging that benefit from their initial spatial positioning. When the whole colony is artificially coated with producer cells, no consumer branches are observed since the mechanism of being initially shoved by producer cells is not possible. These additional simulations confirm our hypothesis of how the consumer branches emerge in silico under weak mutualistic conditions.

We added a description of the importance of cell shoving and initial spatial positioning to the manuscript.

Lines 276 - 287: "An in depth experimental evaluation of the importance of initial spatial positioning and the role of cell shoving is complex due to massive amount of replicates and inability to remove cell shoving in a living community. Alternatively we have removed the shoving component in the mathematical model (i.e. allowing only peripheral cells contribution to colony expansion through placement of their progeny on unoccupied grid nodes). The result was an overall lower expansion of the colonies (SI Figure 1a,b and c) with conspicuous omission of super sectors in the weak mutualistic scenario (SI Figure 1b). Additionally, we could easily control the initial positioning of consumers and producers in the mathematical model. Encasing the colony with consumer cells resulted in myriads of consumer super sectors (SI Figure 1d) whereas encasing the inoculated colony with producer cells suppressed the emergence of any consumer super sectors as they are buried behind the advancing producer front. These additional simulations highlight the importance of initial spatial positioning and cell shoving for the seeding of the observed consumer super sectors in case of the weak mutualistic scenario."

SI Figure 1: Importance of bacterial shoving and initial spatial positioning for pattern formation. Removal of the shoving algorithm (i.e. only cells at the colony periphery can contribute to colony expansion through placing their progeny on unoccupied grid nodes) results in a similar pattern (although reduce in expansion) for the competitive scenario (a), a suppression of super sectors in case of weak mutualism (b) and a significantly reduced expansion rate in the strong mutualistic scenario (c). In case of weak mutualistic conditions, initial positioning of consumer cells is fundamental for colony pattern formation. When the whole periphery is occupied by consumer cells (d), numerous consumer super branches emerge since consumer cells are frequently pushed ahead by the proliferating producer. When the inoculated colony periphery is dominated by producer cells (e), no consumer branches emerge since the consumer is buried behind the advancing producers.

2. In line 174-176+ the authors state "Super sectors that penetrate the layer of producers are typically monoclonal, owing to the fact that they typically nucleate from a single cell" This seems like an important point, making it worth quantifying "typically."

We have quantified the average number of lineages per branch in the simulations and added these data to the manuscript.

Lines 176 - 178: "Super sectors that penetrate the layer of producers are typically monoclonal, owing to the fact that they typically nucleate from a single cell (ratio of consumer lineages at the periphery to consumer branches is 1.25 ± 0.05)."

3. While I enjoyed the section on structured environments, it made me curious how the regular structure and size of these ordered arrays of 'holes' could impact the outcome. As 'lucky' events are important, could homogeneity suppress them more than heterogeneous environments?

This is indeed an interesting point and we have expanded the geometries in the model to include two stochastic domains that vary in pillar radii and density. The results have been added as a visualization to the supplementary material (SI Figure 2) and is shown below for simplicity. The addition of stochastic pillar size results in a heterogeneous pore space similar to the case where the spatial positioning of homogeneous pillars would be chosen stochastically and we thus only created simulations with stochastic pillar size with a homogeneous spacing.

Overall, we find that a structured environment with stochastic pillars created more bottlenecks for bacterial expansion, which results in more coalescence events and less lineages persisting to the colony periphery. In short, the addition of stochastic pillars amplified the findings of the analysis when only using regular grids as a structured environment.

We referred to these simulations in the manuscript in the discussion section.

Lines 344 - 347: "Because natural habitats are rarely homogeneous (as growth on agar plates and in some of the simulations), we explored scenarios with stochastic variations in the size and spacing of obstacles (solid grains or pillars) that showed a similar trend when compared to the simulation results with regular spaced grids (SI Figure 2, SI Table 1)."

SI Figure 2: Inclusion of stochastic variations in particle size simulations of structured habitats with three different obstacle scaping. The inclusion of additional heterogeneity through stochastic variation of obstacle diameter whilst keeping the spacing equal results in an amplified effect of lineage coalescence for all scenarios and a reduction in observed consumer branches in the weak and strong mutualistic scenarios.

SI Table 1: Quantification of total number of peripheral lineages [#], persistent consumer branches [#] and biomass [10^{-8} kg] for competition (CT), weak mutualism (WM) and strong mutualism (SM) in all simulated habitat scenarios. Table shows mean values of all replicates including standard deviations in brackets.

	Total lineages [#]			Consumer branches [#]			Biomass [kg]		
	CT	WM	SM	CT	WM	SM	CT	WM	SM
Homogeneous	163 (13)	143 (12)	283 (12)	85 (7)	22 (6)	110 (5)	3.43 (0.05)	3.18 (0.10)	1.74 (0.01)
Regular coarse	107 (3)	89 (7)	173 (4)	52 (2)	27 (6)	68 (5)	2.96 (0.02)	2.70 (0.06)	1.56 (0.01)
Regular medium	87 (6)	73 (6)	134 (18)	44 (6)	19 (3)	51 (8)	2.56 (0.24)	2.31 (0.22)	1.37 (0.02)
Regular fine	68 (4)	65 (5)	100 (5)	33 (6)	14 (4)	36 (3)	1.91 (0.03)	1.71 (0.03)	1.03 (0.01)
Stochastic coarse	100 (5)	86 (8)	175 (8)	49 (4)	25 (8)	66 (4)	2.95 (0.03)	2.69 (0.06)	1.55 (0.01)
Stochastic medium	85 (3)	73 (6)	143 (7)	40 (5)	23 (6)	54 (4)	2.64 (0.03)	2.37 (0.03)	1.36 (0.01)
Stochastic fine	62 (4)	56 (3)	98 (3)	30 (5)	15 (2)	38 (5)	1.85 (0.05)	1.62 (0.04)	1.00 (0.02)

4. I found figure 4 difficult to interpret. It appears the plots on the bottom row are associated with A, B, and C? This would be clearer if all panels were labeled A-F. Further, I am interested in seeing all quantities measured for all scenarios.

We have relabelled the panels from a) through to f) for clarification and updated the Figure legend and manuscript text where necessary. In addition, we added a table to the supplementary information that includes all the quantifications for all scenarios including the stochastic particle size variation.

Minor comments:

1. It's a small detail, but I did not see the size or spacing of the 'holes' for the structured environment reported. This information would be important for someone to replicate the simulations.

As it is an important omission we have added the description to the methods section for all scenarios.

Lines 451 - 456: "The physical obstacles (in the structured scenarios) are treated as solid and impenetrable domain boundaries. The structured habitats were simulated by adding solid obstacles of defined dimensions (100 microns diameter) spaced regularly within the domain (500 microns, 400 microns and 300 microns for the coarse, medium and fine structured scenarios). Stochastic obstacle simulations in SI Figure 2 contained the medium and fine spacing, with obstacle diameters taken from a normal distribution with mean 100 microns and variance 20 microns."

2. Experiments testing the structured environment simulations would be very interesting if possible, but are not required for publication, in my opinion.

Indeed, we would ourselves be interested in such experiments. In a similar experimental setup when compared to our simulation domains, Ciccarese et al. have performed experiments in structured environments to observe the effect of spatial structure on self-organization dynamics³. Experiments that have controllable boundary conditions similar to the mathematical model would be intricate to perform (the actual physical domain, boundary conditions and visualization). We have included a more prominent reference to the above-mentioned paper in the manuscript.

Lines 334 - 336: "Specifically, a recent study investigated the growth and spatial self-organization of trophically interacting synthetic communities during range expansion with the inclusion of a spatially structured habitat³."

3. It may be worth citing Lowery, Vallespir and Ursell "Structured environments fundamentally alter dynamics and stability of ecological communities." PNAS (2019). While this paper does not investigate range expansions, it does study the effect of spatial structure on competition.

We thank the reviewer for pointing towards this publication.

Lines 331 - 332: "Bacterial assemblages rarely grow on homogeneous surfaces and the spatial context has been shown to profoundly influence the community stability and function with respect to trophic interactions⁴."

4. This is purely the authors' choice, but the title primarily refers to one of the three scenarios investigated here. While it is, to me, the most interesting of the three, I was surprised that the paper was more balanced between the three cases than the title might suggest.

To better represent all trophic interactions and resulting spatial organizations in response to metabolic landscape feedbacks we have changed the title to:

"Spatial organization in microbial range expansion emerging from trophic dependencies and successful lineages"

Reviewer #2 (Remarks to the Author):

In this manuscript, the authors used agent-based mathematical model to tracks individual cell lineages and more interestingly explored the range expansion interrupted by physical obstacles. Combining experiments and model to shape the community patterns during bacterial range expansion is of importance and not a trivial effort. Also, attempt to identify the lineage of community pattern is worthwhile. However, I am not convinced of the value of the model, as explained below.

We would like to thank reviewer 2 for the constructive comment. We included additional information concerning the 3D structure of colonies to the manuscript and how boundary conditions are handled in the structured habitat scenarios.

1. The insight gained in this study was based on a 2-dimensional model. However, the colony growth is a typically 3-dimensional structure formation process. Bacterial cells grow/expand vertically and horizontally. The distributions of bacterial cells and nutrient in the colony at Z-axis have been recently discussed. e.g., eLife 8: e41093 (2019). The main results including lineage tracking and habitat structure might be very different if a 3D model was applied.

Indeed, the dimensionality question raised by the reviewer is a very important one and there is no simple experimental method for testing the outcomes for the colony size in this manuscript. Single cell resolution tracking and the visualization of three-dimensional structures in bacterial micro colonies has been achieved recently⁵, however the tracking capabilities do not extend to colonies the size of our experimental system. Furthermore, the primary region of growth and nutrient penetration is restricted to base of the colony (approximately 10 microns)⁶, which is also the part of the colony that primarily influences the underlying nutrient landscape⁷. Our mathematical model does not resolve the physics of bacterial shoving at the single cell level due to the representation of bacterial cells as super-agents. Although incorporating a three-dimensional shoving algorithm may give further insights into the internal structure of the colony, these results may be difficult to validate due to the coarse representation of three-dimensional shoving and lack of experimental data and thus remain speculative.

2. In the agent based model, the authors used a simple ongrid shoving algorithm to model the colony expansion. However, the actual shoving process is a stochastic one. How would this affect the lineage tracking? Besides, in the presence of obstacles, the authors didn't clearly explain how they handled the boundary conditions where the cells crash to the obstacles. I would expect the types of boundary conditions can result different cell lineage structures in space.

Overall, the shoving algorithm includes two stochastic elements. Firstly, although the shortest path between the cell and the colony periphery is used for shoving, there are often more than one path along the grid, which are of the same shortest length. Under those conditions, a shortest path is chosen at random between all variants. Secondly, the cell residing at the colony periphery is allocated an adjacent, unoccupied grid node at random which may alter the proliferation of different lineages.

In the event that a cell collides with the domain boundary or a pillar in the structured habitat, we added clarification in the materials and methods section.

Lines 451 - 452: "The physical obstacles (in the structured scenarios) are treated as solid and impenetrable domain boundaries."

Minor points:

I found that figure 2ab is rather difficult to understand.

For clarification, we split nitrate and nitrite in panel 2b and added further explanation to the Figure caption.

Reviewer #3 (Remarks to the Author):

In manuscript COMMSBIO-20-0997-T the range expansion of surface-attached cross-feeding bacterial communities is studied under different interaction scenarios, going from competition for two complete reducer strains to weak or strong mutualism for two complementary strains, a producer that reduces nitrates to nitrite, and a consumer that uses nitrite. The level of mutualism is controlled by the pH of the medium in the experiments, which controls the toxicity of the nitrite. Low toxicity for high pH implies weak mutualism, while high toxicity for low pH induces strong mutualism as the producer can not live without the consumer. This interaction, mediated by the diffusion of nitrate and nitrite in space induces self-organization in space, displaying different patterns in each case. They study also the effect of a spatial structure in the expansion of the colony. They find that structured space reduce in general the number of cell lineages because some get blocked at the obstacles.

This work reports a detailed analysis including experimental and numerical results. The results are interesting and convincing, and shade light on the role of the interaction and spatial inhomogeneities in shaping bacterial communities. The article is also clearly written and reasonable easy to follow by a broad audience. There are however a few point (some minor) that need clarification before I can recommend the manuscript for publication in Nature Communications Biology.

We would like to thank the reviewer for the thoughtful and constructive comments. Since some issues raised are similar to those of reviewer 1, we have referred to those descriptions further above.

1. In my opinion the title of this work, if catchy, is not informative enough. I would say that the fact that some lineages proliferate by change is not the main result of the paper, but the identification of the patterns emerging for each different type of interaction and spatial structure is. I would suggest the title to reflect the more general results and not focus only on the "lucky individuals".

In order to better represent all trophic interactions and resulting spatial organizations in response to metabolic landscape feedbacks we have changed the title to:

"Spatial organization in microbial range expansion emerging from trophic dependencies and successful lineages"

2. At some point is not clear to me what is new and what is not with respect to previous literature. Is the model new? Is it the technique to control the interaction type with the pH? The introduction of spatial structures? This should be clearly explained in the text.

We have highlighted more prominently that the main novelty of this manuscript is the mechanism of feedbacks between the emerging colony spatial organization and self-engineered nutrient landscapes depending on trophic interactions.

Lines 66 - 69: "Our modelling framework provides novel mechanistic insights into experimental observations of spatial self-organization during bacterial range expansion, predicting a feedback between the emerging colony pattern and self-engineered nutrient landscape at a resolution currently inaccessible to experimental quantification."

Lines 302 - 306: "Although the resulting colony pattern in our system is similar to the above-mentioned studies, where growth rate differences are imposed via mutations and are thus fixed in time, the mechanism is fundamentally different. In our simulations however, the emerging spatial organization of the synthetic community is a feedback of self-engineered nutrient landscapes that drive localized growth conditions."

We also clarified that the experimental setup and method of tuning trophic interactions has been established previously and is not a novelty of this study.

*Lines 373 - 374: "A previously established synthetic cross-feeding bacterial ecosystem composed of two isogenic mutant strains of the bacterium *Pseudomonas stutzeri* A1501^{8,9} was used as a model community (Fig. 1)."*

3. Only periodic arrays of obstacles are considered. How would the results change if the obstacles were random in size as well as their spatial distribution? By the way, how the spatial inhomogeneities are implemented in the model is not explained.

We have added simulations with stochastic distributions of the different pillar geometries as supplementary information. Details of our findings are outlined in the answer to reviewer 1 major comment 3 above.

4. Concerning the model, it is not explained how the consumption of nutrients, R in Eq. (1), depends on the biomass. I understand it through Eq. (6) but it could be explicitly written. The creation of nitrite is mentioned after Eq. (6), but I would prefer to see the set of coupled differential equations for the concentrations of nitrate, nitrite and biomass of each strain explicitly written. Also how the source of nitrate is included in the equations should be mentioned.

We have expanded Equation 1 to include a description of how nitrate and nitrite are consumed and produced by the two strains in the form of a coupled system of differential equations. Since the strains obey different kinetic laws depending on the trophic interaction, listing equations for all strains is impractical and would result in a large table of differential equations. We have added information on the nitrate source for the simulations.

Lines 414 - 417: "A constant peripheral source for nitrate is included (setting the concentration at the boundary nodes to 1 mM before the numerical diffusion step) whereas nitrite is produced locally solely by bacterial metabolism."

5. The authors consider one dimensional diffusion between nodes on a hexagonal lattice. This is a little strange to me. I would find more natural to implement a two dimensional diffusion on the hexagonal lattice. This is not a minor difference, as considering the one dimensional diffusion between nodes does not properly implement the role of the curvature of concentration fronts in the dynamics.

We agree with the reviewer that there is an inherent bias when using pseudo two-dimensional diffusion represented as one-dimensional diffusion between nodes as shown in the Figure below (comparing our algorithm of 1D diffusion on a hexagonal grid with 2D diffusion on a rectangular grid). In this example, diffusion of an arbitrary concentration is calculated for an arbitrary space with unified time (30 time steps). Although there is some slight deviation between the two, the overall spatial distribution and magnitude is very similar. The main advocate for using a hexagonal grid lies in its simultaneous use for the shoving algorithm. Compared to a square grid, a hexagonal grid provides a more natural geometry for the shoving algorithm. Evidently, a two-dimensional diffusion algorithm could be coupled with a hexagonal shoving grid (or off-grid shoving algorithm), however due to the increase in computational complexity and potential numerical inaccuracies (interpolating the concentration experienced by each cell on the hexagonal grid from a rectangular grid) we

concluded that the one-dimensional diffusion algorithm is adequate for this model.

6. The rule for cell division is not explained either.

We have added the omitted information concerning cell division to the manuscript.

Lines 444 - 447: " Upon reaching intrinsic critical mass for division, a bacterial cell divides into two where one daughter cell occupies the current location and the second either occupies an adjacent cell (if the location is unoccupied or cell shoving is permitted) or is added to the current location in a layer above (pseudo three-dimensional colony growth)."

7. There is a typo in line 133.

We have corrected the typo in the manuscript – many thanks.

References:

1. Karig, D. et al. *Stochastic Turing patterns in a synthetic bacterial population. Proc. Natl. Acad. Sci. U. S. A. (2018).*
2. Liu, Q. X. et al. *Phase separation explains a new class of selforganized spatial patterns in ecological systems. Proc. Natl. Acad. Sci. U. S. A. (2013).*
3. Ciccarese, D., Zuidema, A., Merlo, V. & Johnson, D. R. *Interaction-dependent effects of surface structure on microbial spatial self-organization. Philos. Trans. R. Soc. Lond. B. Biol. Sci. (2020).*
4. Lowery, N. V. & Ursell, T. *Structured environments fundamentally alter dynamics and stability of ecological communities. Proc. Natl. Acad. Sci. U. S. A. 16, 379–388 (2019).*
5. Hartmann, R. et al. *Emergence of three-dimensional order and structure in growing biofilms. Nature Physics (2019).*
6. Warren, M. R. et al. *Spatiotemporal establishment of dense bacterial colonies growing on hard agar. Elife (2019).*
7. Cole, J. A., Kohler, L., Hedhli, J. & Luthey-Schulten, Z. *Spatially-resolved metabolic cooperativity within dense bacterial colonies. BMC Syst. Biol. 9, 1–17 (2015).*
8. Lilja, E. E. & Johnson, D. R. *Segregating metabolic processes into different microbial cells accelerates the consumption of inhibitory substrates. ISME J. 10, 1568–1578 (2016).*
9. Goldschmidt, F., Regoes, R. R. & Johnson, D. R. *Successive range expansion promotes diversity and accelerates evolution in spatially structured microbial populations. ISME J. (2017).*